# Early alterations of social brain networks in young children with autism

Holger Franz Sperdin[1†]*, Ana Coito[2†], Nada Kojovic[1], Tonia Anahi Rihs[2], Reem Kais Jan[2,3], Martina Franchini[1], Gijs Plomp[4], Serge Vulliemoz[5], Stephan Eliez[1], Christoph Martin Michel[2], Marie Schaer[1]

[1]Developmental Imaging and Psychopathology Laboratory, Department of Psychiatry, University of Geneva, Geneva, Switzerland; [2]Functional Brain Mapping Laboratory, Department of Fundamental Neurosciences, University of Geneva, Geneva, Switzerland; [3]College of Medicine, Mohammed Bin Rashid University of Medicine and Health Sciences, Dubai, United Arab Emirates; [4]Department of Psychology, University of Fribourg, Fribourg, Switzerland; [5]EEG and Epilepsy Unit, Neurology, University Hospitals of Geneva, Geneva, Switzerland

**Abstract** Social impairments are a hallmark of Autism Spectrum Disorders (ASD), but empirical evidence for early brain network alterations in response to social stimuli is scant in ASD. We recorded the gaze patterns and brain activity of toddlers with ASD and their typically developing peers while they explored dynamic social scenes. Directed functional connectivity analyses based on electrical source imaging revealed frequency specific network atypicalities in the theta and alpha frequency bands, manifesting as alterations in both the driving and the connections from key nodes of the social brain associated with autism. Analyses of brain-behavioural relationships within the ASD group suggested that compensatory mechanisms from dorsomedial frontal, inferior temporal and insular cortical regions were associated with less atypical gaze patterns and lower clinical impairment. Our results provide strong evidence that directed functional connectivity alterations of social brain networks is a core component of atypical brain development at early stages of ASD.

*For correspondence:
holger.sperdin@unige.ch

†These authors contributed equally to this work

Competing interests: The authors declare that no competing interests exist.

## Introduction

Early preferential attention to social cues is a fundamental mechanism that facilitates interactions with other human beings. During the third trimester of gestation, the human foetus is already sensitive to both voices (*DeCasper and Spence, 1986*) and face-like stimuli (*Reid et al., 2017*). Newborns orient to biological motion (*Simion et al., 2008*) and prefer their mothers' voices to those of other females (*DeCasper and Fifer, 1980*). Infants as young as two weeks imitate faces and human gestures (*Meltzoff and Moore, 1977*). The orientation to and interaction with social cues during infancy drives the later acquisition of social communication skills during toddler and preschool years. As function of experience, the repeated exposure leads to the progressive emergence of adaptive interactions with conspecifics. Alongside, the brain develops a network of cerebral regions specialized in understanding the social behaviours of others. This network includes the orbitofrontal and medial prefrontal cortices, the superior temporal cortex, the temporal poles, the amygdala, the precuneus, the temporo-parietal junction, the anterior cingulate cortex (ACC) and the insula (*Brothers, 1990*; *Frith and Frith, 2010*; *Adolphs, 2009*; *Blakemore, 2008*). Collectively, these areas form the *social brain* and are all implicated to some extent in processing social cues and encoding human social behaviours (*Brothers, 1990*; *Frith and Frith, 2010*; *Adolphs, 2009*; *Blakemore, 2008*).

Autism is a life-long lasting, highly prevalent neurodevelopmental disorder that affects core areas of cognitive and adaptive function, communication and social interactions (*Christensen et al., 2016*). A common observation in infants later diagnosed with ASD is the presence of less sensitivity

**eLife digest** Newborns are attracted to voices, faces and social gestures. Paying attention to these social cues in everyday life helps infants and young children learn how to interact with others. During this period of development, a network of connections forms between different parts of the brain that helps children to understand other people's social behaviors.

During their first year of life, infants who later develop autism spectrum disorders (ASD) pay less attention to social cues. This early indifference to these important signals leads to social deficits in children with ASD. They are less able to understand other people's behaviors or engage in typical social interactions. It's not yet clear why children with ASD are less attuned to social cues. But is likely that the development of brain networks essential for understanding social behavior suffers as a result. Studying how such networks develop in typical very young children and those with ASD may help scientist learn more.

Now, Sperdin et al. confirm there are differences in the social brain-networks of very young children with ASD compared with their typical peers. In the experiment, 3-year-old children with ASD and without watched videos of other children playing, while Sperdin et al. recorded what they looked at and what happened in their brains. Eyemovements were measured with a tracker, and the brain activity was recorded using an electroencephalogram (EEG), which uses sensors placed on the scalp to measure electrical signals.

What children with ASD looked at was different than their typical peers, and these differences corresponded with alterations in the brain networks that process social information. Children with ASD who had less severe symptoms had stronger activity in these brain networks. What they looked at also was more similar to typical children. This suggests less severely affected children with ASD may be able to compensate that way.

Identifying ASD-like behaviors and brain differences early in life may help scientists to better understand what causes the condition. It may also help clinicians provide more individualized therapies early in life when the brain is most adaptable. Long-term studies of these brain-network differences in children with ASD are necessary to better understand how therapies can influence these changes.

and diminished preferential attention to social cues during the first year of life (*Osterling and Dawson, 1994*). Toddlers with ASD orient preferentially to non-social contingencies (*Klin et al., 2009*). Indifference to voices (*Sperdin and Schaer, 2016*) and faces (*Grelotti et al., 2002*) in ASD leads to deficits in the development of adapted social interactions with others and to difficulties in understanding human behaviours. It is not established why children with ASD show insensitivity to stimuli with social contingencies at early developmental stages, but this apparent indifference to social cues ultimately hinders the normal development of the *social brain* network or parts thereof (*Pelphrey et al., 2011*; *Gotts et al., 2012*). Some authors propose that deficits in the development of social cognition (such as learning to attribute mental states to others, 'theory of mind' [*Frith, 1989*]) and/or in sensory processing (*Dinstein et al., 2012*) prevent children with ASD to actively and appropriately engage with social stimuli. Another hypothesis suggests that they have difficulties building up stimulus-reward contingencies for social stimuli, due to a reduced motivation to attend and engage with them. Regardless of the reasons behind reduced social orienting, diminished interaction and exposure to social stimuli may in turn impede the development of the *social brain* at early developmental stages in ASD (*Chevallier et al., 2012*; *Dawson et al., 2004*).

Evidence remains limited for brain network alterations in response to socially meaningful stimuli in ASD during the period spanning the toddler to preschool years, partly because the acquisition of data during that age period is extremely challenging (*Raschle et al., 2012*). However, studying very young children with ASD, closer to their diagnosis, is all the more important when recent findings suggest the presence of major developmental changes in large-scale brain networks between adults and younger individuals with ASD (*Nomi and Uddin, 2015*). Currently, it remains unclear how autism affects the development of the functional brain networks implicated in the processing of socially meaningful information at early developmental stages. A better delineation of the timing and nature

of the neurodevelopmental alterations associated with core social deficits in autism may in turn help to improve therapeutic strategies.

Electroencephalography (EEG) is as a powerful non-invasive method to study atypical brain responses to social stimuli in clinical paediatric populations with ASD. For example, surface-based experiments have reported aberrant evoked potentials in response to dynamic eye gaze in infants at high-risk for ASD (*Elsabbagh et al., 2012*) or to speech stimuli in toddlers with ASD (*Kuhl et al., 2013*) with differences in resting EEG power in infants at high-risk for ASD (*Tierney et al., 2012*). Whilst useful, most of the EEG experiments performed on very young children with ASD (younger than four years) have been done with few electrodes only and the analysis restricted to the sensor space. Therefore, hypothetical alterations in the functional brain networks underlying the processing of social stimuli remain to be determined for that age period in ASD.

Here, we recorded high-density EEG and high resolution eye-tracking in toddlers and pre-schoolers with ASD and their TD peers as they watched naturalistic and ecologically valid dynamic social movies. Using data-driven methods, we first investigated whether the visual exploration behaviour was atypical in toddlers and preschoolers with ASD using kernel density distribution estimations. Then, we explored whether their ongoing source-space directed functional connectivity was altered compared to their TD peers using Granger-causal modelling applied to the EEG source signals. This method estimates brain connectivity in the frequency domain. It identifies which brain regions are the key drivers of information flow in a brain network and directional relationships between brain regions that belong to a network (*Coito et al., 2016b*). This approach has been

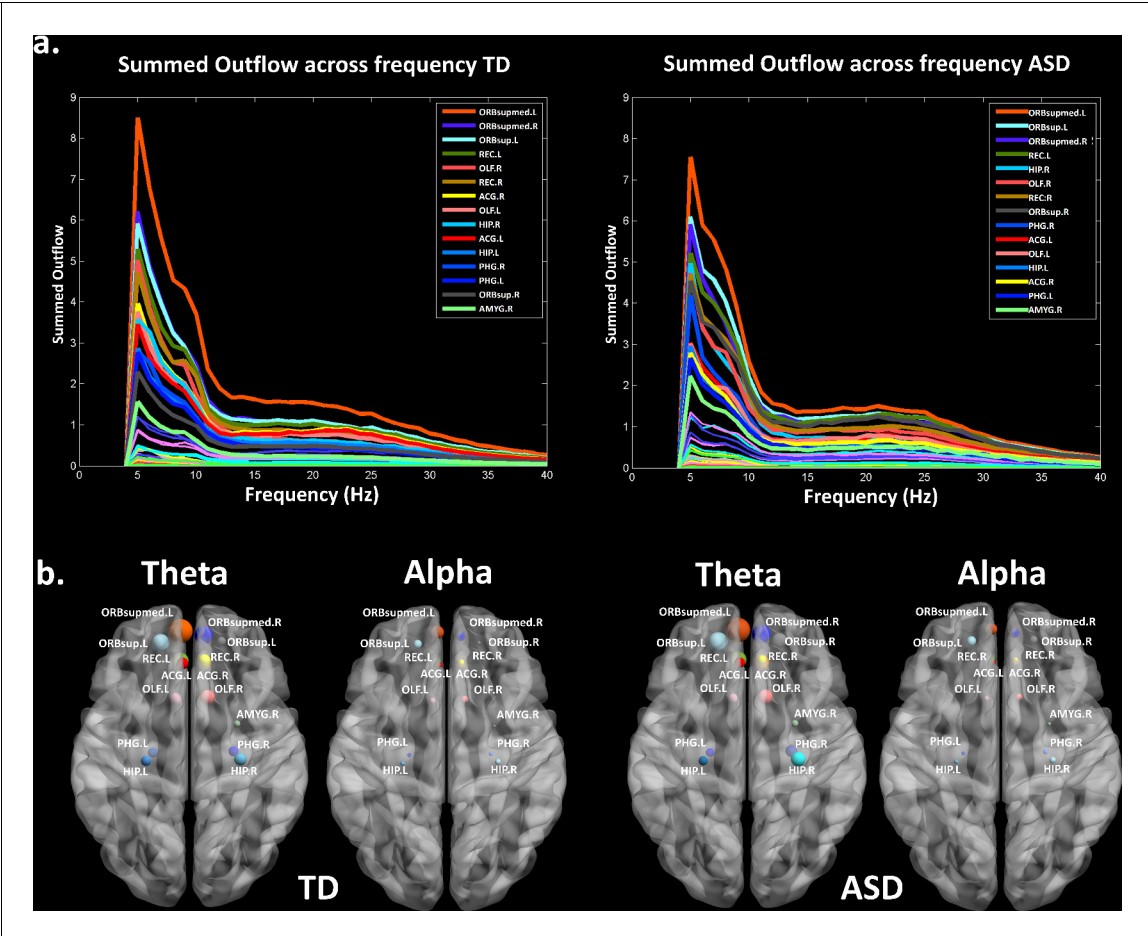

**Figure 1.** Summed outflow of the largest drivers across frequencies. (**a**) The summed outflow of the largest drivers across frequencies is illustrated for each group (TD, Left; ASD, Right). (**b**) Regions consistently showing large driving in both groups for theta and alpha. Summed outflows are represented as spheres: the larger the sphere, the higher the summed outflow. ROIs are displayed on an ICBM Average Brain, axial top view. See acronyms list in *Table 2*.

applied to study connectivity alterations using intra-cranial recordings (*Wilke et al., 2009*; *van Mierlo et al., 2013*; *van Mierlo et al., 2011*) as well as source imaging based on EEG recordings in clinical populations (*Ding et al., 2007*; *Coito et al., 2016a*; *Coito et al., 2015*; *Coito et al., 2016b*) and in healthy human participants (*Astolfi et al., 2007*; *Hu et al., 2012*; *Plomp et al., 2015b*). Finally, we looked for relationships between directed functional connectivity measures, visual exploration behaviour and clinical phenotype. As toddlers with ASD have less preferential attention for social cues, we hypothesized that they would show both a different visual exploration behaviour of the dynamic social images and altered directed functional connectivity patterns in brain regions involved in processing social information compared to their TD peers.

## Results

### Summed outflow

The summed outflow (i.e.- the amount of information transfer) is a measure that reflects the importance (i.e.-the amount of driving) of a given region of interest (ROI) in the network (see Materials and methods section). To understand the functional wiring and the dynamic flow underlying the processing of the dynamic social stimuli, we used a data-driven method to explore in which frequency band the highest summed outflow occurred in 82 ROIs across the whole brain. A ROI with a strong summed outflow has a key role in directing the activity towards other ROIs in the network. The strongest summed outflow across the whole brain occurred in the theta band (4–7 Hz) in both groups. The summed outflow of the largest drivers across frequencies is illustrated for each group in *Figure 1a*. As can be seen, the largest peaks of activity are present in the theta band range (4–7 Hz) in both groups followed by peaks of activity in the alpha band range (8–12 Hz). The global driving in the theta and alpha bands did not differ between groups (theta : df = 34, t = 0.6201, p = 0.536; alpha : df = 34, t = 0.1736, p = 0.8632). Driving in the theta band was higher compared to the driving in the alpha band in both groups (ASD: df = 17, t = 11.86, p < 0.0001; TD:df = 17, t = 8.025, p < 0.0001). Several regions common to both groups showed a large driving (summed outflow) in both frequency bands, and notably the bilateral medial frontal and superior orbitofrontal regions, the bilateral hippocampi, the bilateral ACC and the right amygdala (*Figure 1b*).

Thereafter, we characterized the differences in the summed outflow across all brain regions between the groups in the theta band and in the alpha band, separately. For the theta band, we identified six ROIs that showed a statistically higher driving (stronger summed outflow) in the ASD group in comparison to the TD group (Mann-Whitney-Wilcoxon test, two-tailed, p<0.05): the right orbital part of the superior frontal gyrus (Ws = 267, z = −2.088. p=0.037, r = −0.348), the bilateral orbital parts of the middle frontal gyri (Left: Ws = 259, z = −2.341, p=0.019, r = −0.39; Right: Ws = 252, z = −2.563, p=0.01, r = −0.427), the right middle cingulate gyrus (Ws = 259, z = −2.341, p=0.019, r = −0.390), the left superior occipital gyrus (Ws = 270, z = - 1.993, p=0.047, r = −0.332), and the left superior temporal gyrus (STG) (Ws = 255, z = −2.468, p=0.013, r = −0.411) (*Figure 2a*). This indicates the presence of a stronger driving from these regions in the toddlers and preschoolers with ASD compared to their TD peers when viewing the dynamic social images. For the alpha band, we identified three ROIs that had a different driving in the ASD group in comparison to the TD group (Mann-Whitney-Wilcoxon test, two-tailed, p<0.05). The the right orbital part of the middle frontal gyrus (Ws = 262, z = −2.246, p=0.024, r = −0.374) and the left cuneus (Ws = 265, z = −2.151, p=0.031, r = −0.358) had a higher driving and the right STG had a weaker driving (Ws = 265, z = −2.151, p=0.031, r = −0.358) (*Figure 2a*). The boxplots with the summed outflow values for each group and for each of the significant ROIs are displayed in *Figure 2b* for theta and *Figure 2c* for alpha.

### Region-to-region directed functional connectivity

We looked for differences in the region-to region directed functional connectivity using Granger-causal modelling (see Materials and methods section) from each of the six nodes for the theta band, and from each of the three nodes in the alpha band separately in both groups. In the theta band, all the connections from the six ROIs in the toddlers and preschoolers with ASD were stronger than the strongest connections in the TD participants (Mann − Whitney − Wilcoxon, two − tailed, p < 0.05, Benjamini − Hochberg = 0.05). This suggests the presence of hyper-connectivity in the toddlers and

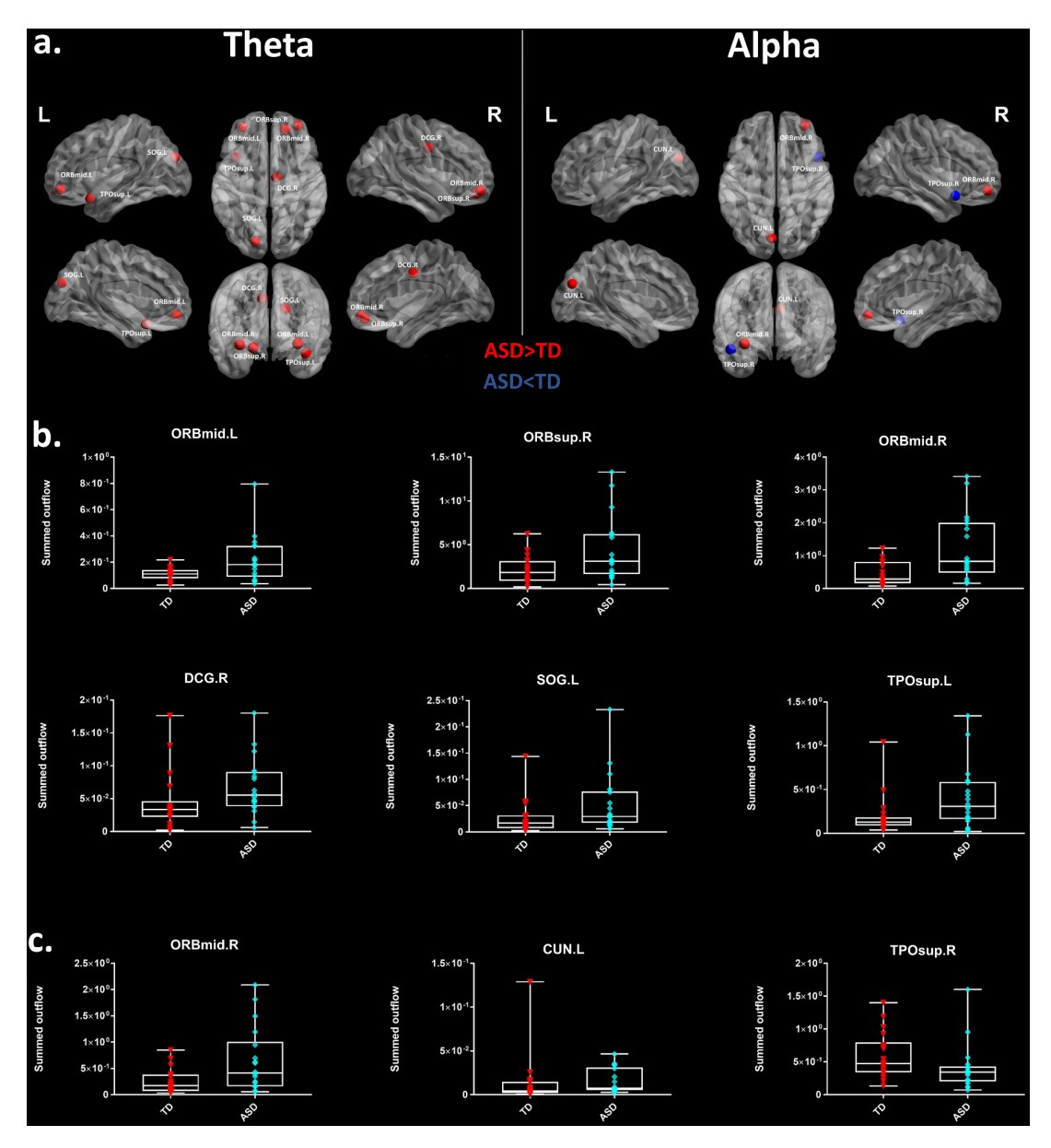

**Figure 2.** Summed outflow differences in the ASD group compared to their typically developing peers. (**a**) On the left, the 6 ROIs with a statistically significant different summed outflow in the ASD group compared to their TD peers for the theta band. On the right, the 3 ROIs with a statistically significant different summed outflow in the ASD group compared to their TD peers for the alpha band. A red nod indicates increased driving, a blue nod indicates decreased driving. Corresponding ROIs are displayed on an ICBM Average Brain, with sagital, axial and corronal views. (**b**) Boxplots with the summed outflow values comparing each group for each significant ROI in the theta band. (**c**) Boxplots with the summed outflow values comparing each group for each significant ROI in the alpha band. The boxplots display the full range of variation of the summed outflows (from min to max), rectangles span the interquartile range and the median. See acronyms list in *Table 2*.

preschoolers with ASD in theta. The region-to-region directed functional connectivity from the six ROIs in theta is illustrated in *Figure 3*. The estimation of the region-to-region directed connectivity (i.e. to which other ROIs the activity was directed) also revealed different network patterns for all the six ROIs in the toddlers and preschoolers with ASD compared to their TD peers. The boxplots of the outflow values of the connections from the right orbital part of the superior frontal gyrus seed region are provided in *Figure 3—figure supplement 2* for the ASD group and *Figure 3—figure supplement 2* for the TD group. In the alpha band, the region-to-region directed functional connectivity

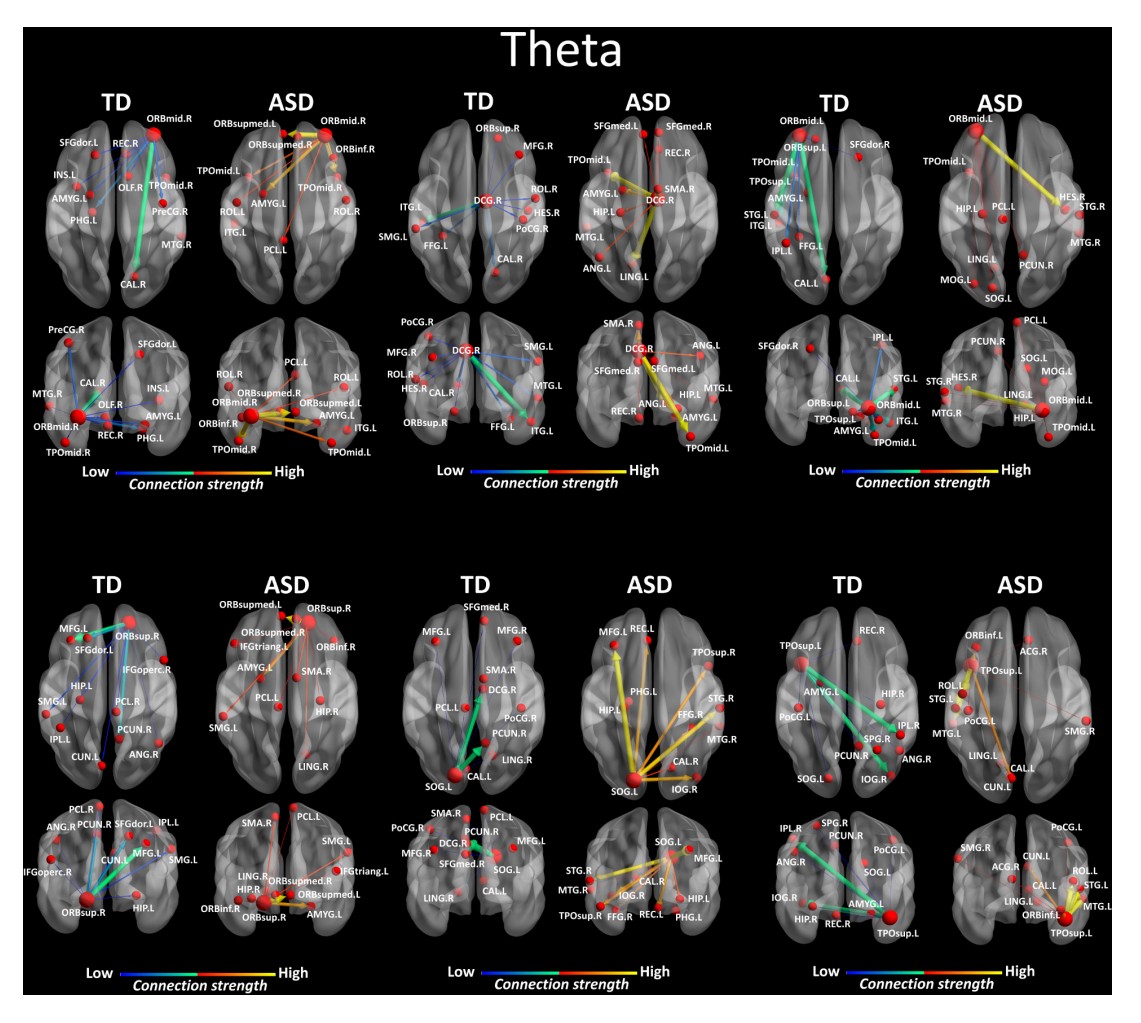

**Figure 3.** Region-to-region directed functional connectivity for the theta band (4–7 Hz) from each of the six significant ROIs represented as large red spheres. Outflows are represented as arrows: the larger the arrow, the stronger the outflow. ROIs and connections are displayed on an ICBM Average Brain, axial and coronal views. See acronyms list in *Table 2*.

The online version of this article includes the following figure supplement(s) for figure 3:

**Figure supplement 1.** ORBsup.R seed (large red sphere) for the ASD group in theta band.

**Figure supplement 2.** ORBsup.R seed for the (large red sphere) for the TD group in theta band.

analysis revealed stronger connections from the right orbital part of the middle frontal gyrus and the left cuneus whereas the right STG had weaker connection in the toddlers and preschoolers with ASD compared to their TD peers (Mann-Whitney-Wilcoxon, two-tailed, p<0.05, Benjamini-Hochberg = 0.05). Similarly to what we found in the theta band, all three significant ROIs in the alpha band had different network patterns in the toddlers and preschoolers with ASD compared to their TD peers (*Figure 4*).

## Correlations with ADOS-2, PEP-3, VABS-II and gaze Proximity Index

We further explored associations between the summed outflow in the theta and alpha bands from the ROIs and clinical and behavioural phenotypes (Spearman − rho, two − tailed, p <0.05, Benjamini − Hochberg = 0.05). None of the correlations between the summed outflow and ADOS-2 severity scores survived False discovery rate (FDR) correction for either bands (Benjamini − Hochberg = 0.05). For the summed outflow in the theta band, we found strong positive correlations between the summed outflow in the right lingual area and standard scores from the socialization domain (rs =

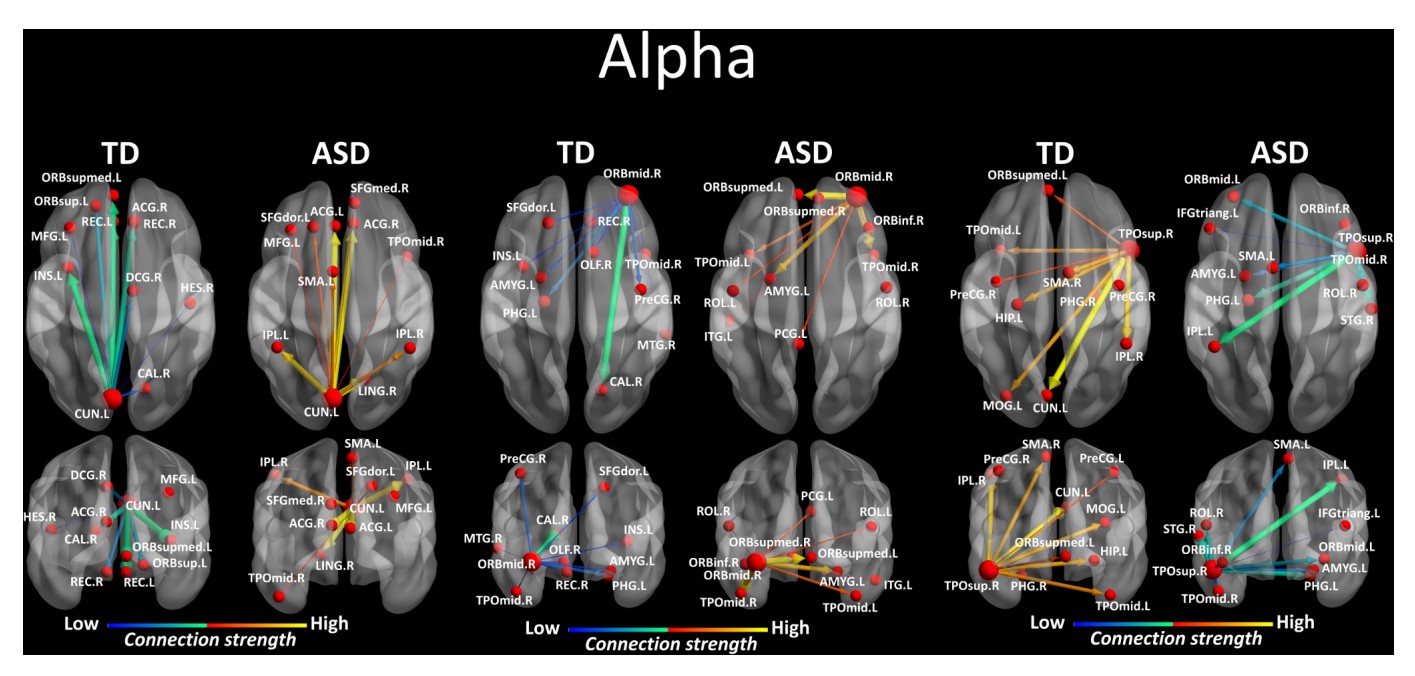

**Figure 4.** Region-to-region directed functional connectivity for the alpha band (8–12 Hz) from each of the three significant ROIs, represented as large red spheres. Outflows are represented as arrows: the larger the arrow, the stronger the outflow. ROIs and connections are displayed on an ICBM Average Brain, axial and coronal views. See acronyms list in *Table 2*.

0.751, N = 18, p=0.0003, two-tailed, <0.05; Benjamini-Hochberg = 0.05) as well as with standard scores from the leisure and play skills subdomain of the VABS-II (rs = 0.802, N = 18, p=0.0001, two-tailed, <0.05; Benjamini-Hochberg = 0.05). None of the correlations between the summed outflow and VABS-II standard scores survived FDR correction for the alpha band (Benjamini-Hochberg = 0.05). Higher summed outflow within the left Heschl area near the posterior convolutions of the insula and the left rolandic operculum near the circular sulcus of the insula rostrally were positively related to better fine (rs = 0.745, N = 18, p=0.0004, two-tailed, <0.05; Benjamini-Hochberg = 0.05) and gross motor skills (rs = 0.744, N = 18, p=0.0004, two-tailed, <0.05; Benjamini-Hochberg = 0.05) as measured by the PEP-3. For the alpha band, higher summed outflow within the left hippocampus and the left rolandic operculum were positively related to better fine (rs = 0.736, N = 18, p=0.0005, two-tailed, <0.05; Benjamini-Hochberg = 0.05) and gross motor skills (rs = 0.737, N = 18, p=0.0005, two-tailed, <0.05; Benjamini-Hochberg = 0.05) as measured by the PEP-3. The toddlers and pre-schoolers with ASD with a gaze pattern similar to their TD peers showed an increased driving in theta within the left middle cingulate cortex (rs = 0.726, N = 18, p=0.0007, two-tailed, <0.05; Benjamini-Hochberg = 0.05) and the right paracentral lobule (rs = 0.738, p = 0.0005, two − tailed, <0.05; Benjamini − Hochberg = 0.05). There was no significant relationship between the Proximity Index (see Materials and methods section) and the summed outflow in the alpha band after FDR correction. The significant correlations between the summed outflows and the Proximity Index, VABS-II standard scores and PEP-3 standard scores for each frequency band are displayed in *Figure 5*. Finally, we explored associations between gaze performance with developmental scores obtained from the PEP-3 and with adaptive scores obtained from the VABS-II (D'Agostino-Pearson omnibus normality test, K2, p<0.05; Pearson's r, two-tailed, p<0.05). We didn't find any significant correlations between the Proximity Index and the global level of autistic severity as measured with the calibrated ADOS-2 severity score. However, we found that the toddlers and preschoolers with ASD with a better gaze performance had better global adaptive functioning as measured by the VABS-II (K2 = 3.339, p=0.1883; r = 0.578, p=0.012), which was driven by better global (K2 = 0.8179, p=0.6643; r = 0.575, p=0.013) and fine (K2 = 0.5438, p=0.7619; r = 0.509, p=0.031) motor skills, and better development of interpersonal relationships (K2 = 5.308, p=0.0704; r = 0.581, p=0.011). We also found

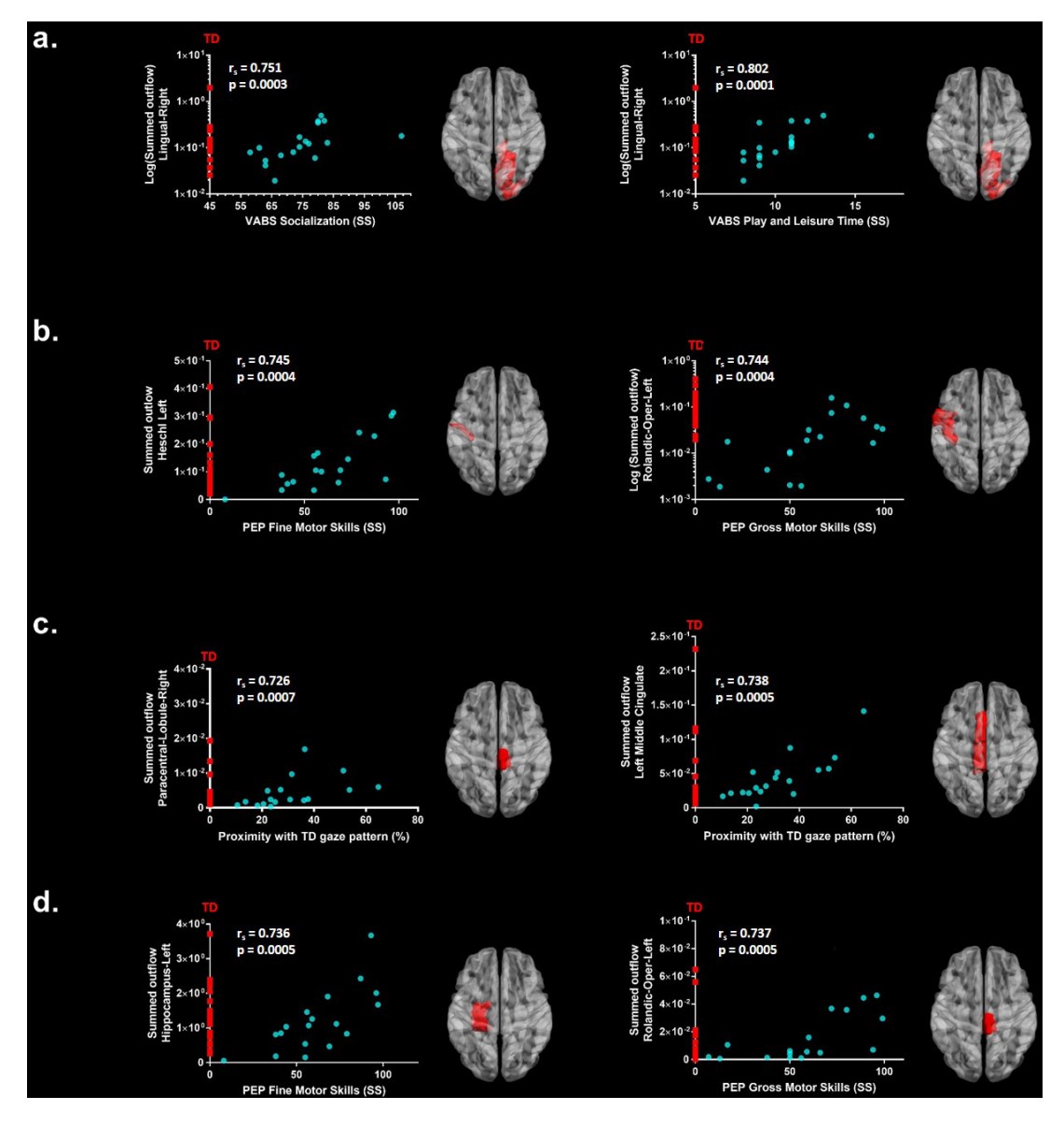

**Figure 5.** Significant correlations after FDR correction between the summed outflow in theta for the ASD participants (represented as blue dots) and (**a**) standardized VABS-II scores, (**b**) PEP-3 standardized scores and (**c**) Proximity Index. (**d**) Significant correlations after FDR correction between the summed outflow in alpha and PEP-3 standardized scores. TD summed outflow values are plotted on the Y axis in red. Corresponding ROI are displayed on an ICBM Average Brain, axial top view.

that these children with a better gaze performance had better visual motor imitation skills (K2 = 2.671, p=0.263; r = 0.534, p=0.022) as measured by the PEP-3.

## Discussion

Abnormal processing of social cues is a hallmark of ASD (*Chevallier et al., 2012*; *Dawson et al., 2004*; *Dichter et al., 2009*; *Elsabbagh et al., 2012*; *Gotts et al., 2012*; *Greene et al., 2011*; *Klin et al., 2009*; *Pelphrey et al., 2011*). However, evidence for alterations of *social brain* networks at early stages of ASD is scant. Using data-driven methods, we observed aberrant gaze patterns together with frequency specific alterations in the directed functional connectivity in toddlers and preschoolers with ASD when exploring dynamic social stimuli compared to their TD peers. These

differences manifested as increased driving and hyper-connectivity in the theta frequency band from nodes that include the right orbital part of the superior frontal gyrus, the bilateral orbital parts of the middle frontal gyri, the right middle cingulate gyrus, the left superior occipital gyrus and the left STG. For the alpha band, we found increased driving from the right orbital part of the middle frontal gyrus and the left cuneus and decreased connectivity from the right STG. To the best of our knowledge, this is the first evidence indicating concomitant alterations in the visual exploration of dynamic social images and in the directed functional connectivity involving key nodes of the *social brain* (*Brothers, 1990*; *Frith and Frith, 2010*; *Adolphs, 2009*; *Blakemore, 2008*) at early stages of ASD.

The results indicate that the highest information transfer (i.e. summed outflow) occurs at the global brain level in the theta band (4–7 Hz) followed by the alpha frequency band (8–12 Hz). As such, our data-driven approach reveals high information transfer in physiologically relevant frequency bands given the young age of our participants. These are, namely, prominent brain rhythms during infancy and toddlerhood (*Saby and Marshall, 2012*; *Orekhova et al., 2006*). Throughout development, slow waves modulate attentional brain states, encode specific information and ease communication between neuronal populations (*Lopes da Silva, 2013*). Theta and alpha bands are thought to underlie different cerebral functions, but are closely related (*Klimesch, 1999*). During infancy and early childhood, modulations in alpha band activity have been associated with the progressive development of visual attentional networks and inhibition of task-irrelevant brain areas (*Orekhova et al., 2001*; *Stroganova et al., 1999*), while theta is thought to play a functional role in memory formation, emotional and cognitive functioning (*Saby and Marshall, 2012*; *Orekhova et al., 2006*).

In our experiment, the highest information transfer occurred in the theta band. In very young children, theta modulations have been related to the development of the *social brain*. For example, surface-based EEG experiments in TD infants report enhanced theta power to social versus non-social stimuli at 12 months (*Jones et al., 2015*). Theta increases during attention to social stimulation in infants and preschool aged children (*Orekhova et al., 2006*). Hence, social contingencies modulate theta band activity. Similarly, our results show high information transfer in theta when toddlers and preschoolers visually explore dynamic social stimuli.

The development of attentional processes in young children has been associated with modulations in alpha band activity (*Orekhova et al., 2001*). Conversely, our young participants had to deploy their attentional focus to the dynamic social stimuli. This would explain why high information transfer was also found in the alpha frequency band.

EEG experiments in individuals with ASD show a reduction or an increase in coherence patterns in the theta and/or alpha frequency bands compared to their TD peers at various ages and under different experimental conditions (*Schwartz et al., 2017*). However, most of the available EEG experiments performed on very young children with ASD and analysis thereof were so far restricted to the scalp surface. As a result, information remains limited regarding the presence of frequency specific alterations within brain regions when young children are exposed to social stimuli. Here, our data-driven source-space approach revealed not only high information transfer in the theta and alpha frequency bands, but also, the involvement of the bilateral medial and the superior orbital frontal regions, the bilateral hippocampi, the bilateral ACC and right amygdala. These areas are implicated in processing social cues and encoding human social behaviours (*Brothers, 1990*; *Frith and Frith, 2010*; *Adolphs, 2009*; *Blakemore, 2008*).

Our results further indicate the presence of frequency specific alterations in the driving from several brain areas in the toddlers and preschoolers with ASD compared to their TD peers. In the theta band, we found an overall dominant higher driving within several frontal and the cingulate regions. In TD individuals, theta generates within the frontal cortices or the ACC (*Asada et al., 1999*). In comparison to their TD peers, both these areas develop differently in young toddlers later diagnosed with ASD (*Schumann et al., 2010*). Accordingly, our results raise the possibility that the brain regions generating theta also follow a different development in the toddlers and preschoolers with ASD. For the alpha band, alteration in the driving was evident from the right orbital part of the middle frontal gyrus. Although experiments during the first three years of life are currently sparse, increased alpha-range EEG connectivity over frontal and central electrodes has recently been reported in high-risk infants who were diagnosed with ASD at 36 month (*Orekhova et al., 2014*). A magnetoencephalography (MEG) study performed at rest using a source-space approach reported lower coherence in the theta and alpha bands within parietal and occipital regions but their ASD group only included adolescents (*Ye et al., 2014*). The differences between this specific study and

our could stem from either variations in the age groups (adolescents versus toddlers and pre-schoolers), the stimuli used (grey cross inside a white circle versus dynamic biological visual stimuli) or the methods. More generally, several factors explain discrepancies in brain connectivity results between studies. The type of connectivity measures applied, the approach (surface versus source based), the brain regions analysed and frequency bands examined are variables that influence the results or the age of the participants (*Mohammad-Rezazadeh et al., 2016*).

Frontal and cingulate areas have been implicated in various complex aspects of social cognition, social reward, social perception and social behaviour (*Jonker et al., 2015*; *Apps et al., 2013*). Meta-bolic changes within the medial prefrontal cortex and the cingulate cortex are correlated with social interaction impairments in childhood ASD (*Ohnishi et al., 2000*). Several experiments report struc-tural (*Patriquin et al., 2016*) and functional (*Gotts et al., 2012*; *Patriquin et al., 2016*; *Greene et al., 2011*; *Cheng et al., 2015*) alterations within these brain areas in school aged chil-dren, adolescents and adults with ASD when compared to their TD peers. A recent study described hyper-connectivity within the ACC and bilateral insular cortices in a sample including children aged between seven to 12 years (*Uddin et al., 2013a*). Some authors propose that the two together form the *salience network*, whose role is to direct attention to behaviourally-relevant stimuli (*Menon and Uddin, 2010*). Although we didn't find differences in the driving from the insula compared to the TD peers, there is an increasing number of evidence showing an abnormal development of the *salience network* or components thereof in ASD (*Uddin, 2015*), which may partially explain the limited inter-est for and engagement with social stimuli that is often observed in individuals with ASD and that constitutes a hallmark of the disorder (*Klin et al., 2009*; *Pelphrey et al., 2011*). Accordingly, the toddlers and preschoolers with ASD had a different visual exploration behaviour of the dynamic social stimuli raising the possibility of a reduced interest to visually engage with them. Alternatively, alterations in driving from these regions could partially reflect a reduced motivation to attend and engage with the dynamic social stimuli (*Chevallier et al., 2012*; *Dawson et al., 2004*). The altera-tions in the driving in the alpha band we found here, could also be related to the presence of devel-opmental impairments in attentional networks and/or inhibitory functions (*Keehn et al., 2013*).

We found higher driving in theta from the left superior temporal and occipital gyri. In the alpha band, we found alterations in the driving from a node in the right STG and the left cuneus in the occipital lobe. Those brain areas are implicated in the processing of biological motion, in analysing the intentions of other people's actions and self-reflection (*Pelphrey and Carter, 2008*; *Pelphrey et al., 2005*; *Pelphrey and Morris, 2006*; *Pelphrey et al., 2004*). Our result would sug-gest that the exploration of the dynamic social visual stimuli that contained biological movements led to altered driving from these brain areas in both frequency bands in the toddlers and pre-schoolers with ASD compared to their TD peers.

Overall hyper-connectivity seems prevalent in younger populations (that is, infants at high-risk for ASD, toddlers and preschoolers with ASD) while hypo-connectivity is more observed during adoles-cence and adulthood in ASD (*Uddin et al., 2013b*). Conversely, a developmental shift occurs in brain growth with an initial period of early brain overgrowth followed by normalization sometime during adolescence (*Courchesne et al., 2011*). Accordingly, structural white matter connectivity studies also highlight this shift from higher structural connectivity in very young children with ASD to lower connectivity in older children with ASD (*Hoppenbrouwers et al., 2014*; *Conti et al., 2015*). There-fore, a global higher-driving and hyper-connectivity from key nodes of *the social brain* in the theta frequency band in our ASD group is consistent with reports in the literature when considering the very young age of our participants (around 3 years of age on average). For the alpha frequency band, we found alterations in the driving manifesting as both hyper-connectivity from a frontal and an occipital area and under-connectivity from a node in the superior temporal pole. Hence, we found frequency-specific network alterations with distinct patterns of directed functional connectivity in the toddlers and preschoolers with ASD compared to their TD peers. This result is in line with recent experiments indicating the presence of distinct patterns of hyper- and hypo-connectivity between brain regions functionally defined by neural oscillatory activity in children and adolescents with ASD (*Ye et al., 2014*; *Kitzbichler et al., 2015*; *Datko et al., 2016*).

We further explored associations between the driving in the nodes of the network (that is, the summed outflow) and clinical and behavioural phenotypes for each frequency band. We didn't observe any significant relationships between summed outflow and the ADOS severity scores after FDR correction in either frequency bands. In theta, we observed an increased driving from the

median cingulate cortex and the paracentral lobule in the toddlers and preschoolers with ASD who had a more similar visual exploration pattern to their TD peers. Thus, an improved visual exploration pattern of the dynamic social images was related to increased summed outflow in theta from these regions. Moreover, higher summed outflow from the right lingual area was related to better socialization behaviour and leisure and play skills as measured by the VABS-II. Higher summed outflow from the left Heschl's area near the posterior convolutions of the insula and the left rolandic operculum near the circular sulcus of the insula rostrally were positively related to better fine and gross motor skills as measured by the PEP-3. Finally, for the alpha band, we found that higher driving within the left hippocampal area and the left rolandic operculum were positively related to better fine and gross motor skills as measured by the PEP-3. As such, overall increased activity in the theta band within dorsomedial frontal, inferior temporal and insular cortical regions were associated with lower clinical impairment and less atypical gaze patterns whereas increased driving in the alpha band was selectively associated with better motor performance. The presence of hyperactivity within relevant brain region has been interpreted as a possible compensatory mechanism when performing a social target detection task, in adults with ASD at least (*Dichter et al., 2009*). While to the best of our knowledge, there is currently no other relevant experimental data that addresses this question, we speculate that the overall hyper-driving from these relevant brain regions might be a mechanism to compensate for atypical development of the brain's circuitry over time as higher directed functional connectivity was related improved socialization, motor behaviours and better visual exploration of dynamic social images. However, longitudinal measurements are necessary to fully confirm this interpretation. De facto, the toddlers and preschoolers with ASD who had better gaze performance had better adaptive behaviour, improved global and fine motor skills and enriched interpersonal relationships as measured by the VABS-II and better visual motor imitation skills as measured by the PEP-3. They were also those who had higher summed outflow in several relevant brain areas.

Beyond functional and structural brain alterations reported elsewhere in older children and adults with ASD, our results suggest for the first time, the presence of frequency specific alterations in the driving of information flow from brain areas implicated in social information processing during the viewing of naturalistic dynamic social images in toddlers and preschool with ASD. Furthermore, we show that these frequency specific directed functional connectivity network alterations within regions of the *social brain* are present at early stages of ASD, justifying further investigation into how early therapeutic interventions targeting social orienting skills may help to remediate *social brain* development during this critical age period when plasticity is still possible. Longitudinal experiments on very young children with ASD are critically needed to better delineate modulations in brain patterns at the time of diagnosis, and how these alterations are influenced by therapeutic intervention. The present experiment is a first step towards that direction.

## Materials and methods

### Participants

Recruitment of toddlers and preschoolers with ASD was achieved via clinical centres specialized in ASD and French-speaking parent associations. TD toddlers and preschoolers were recruited via announcements in the Geneva community. Prior to the experiments, all the procedures were approved by the Ethics Committee of the Faculty of Medicine of the University of Geneva Hospital in accordance with the ethical standards proclaimed in the Declaration of Helsinki. For all participants, an interview over the phone and a medical developmental history questionnaire were completed before their initial visit. All participants' parents gave their informed consent prior to inclusion in the study. 120 participants were recruited for the experiment. We did not manage to put the EEG cap on the head of 23 ASD and 7 TD participants. We managed to put the cap on 90 participants. Out of those, we excluded 28 ASD and 26 TD participants because of too many movement-related artefacts, unrepairable noisy signal, lack of interest, or insufficient amounts of epochs available for subsequent analysis. This was to be expected given the extremely sensitive population at study here. As a result, 36 participants were included: 18 young children with ASD (2 females; mean age 3.1 years ± 0.8, range 2.2–4.4) and 18 age matched (df = 34, t = 2.72, p=0.852) TD peers (5 females; mean age 3.1 years ± 0.9, range 2.0–4.8). All participants with ASD included in the study received a clinical diagnosis prior to their inclusion in the research protocol. Diagnosis of ASD was rigorously

**Table 1.** Characteristics of Study Participants.

| Characteristic | Autism spectrum disorder | Typically developing | | | |
|---|---|---|---|---|---|
| Gender ratio (M/F) | 16/2 | 13/5 | | | |
| | Mean, SD, N | Mean, SD, N | T value | df | P value |
| Age in years | 3.1, 0.8, 18 | 3.1, 0.9, 18 | 0.165 | 34 | 0.87 |
| ADOS CSS | 7.9, 1.6, 18 | 1.1, 0.47, 18 | 17.87 | 34 | 0.000 |
| PEP-Cognitive verbal/pre-verbal | 67.78, 18.85, 18 | 95.81, 7, 16 | −5.87 | 32 | 0.000 |
| PEP-Expressive language | 50.28, 27.49, 18 | 92.94, 8.61, 16 | −6.24 | 32 | 0.000 |
| PEP-Receptive language | 60.06, 23.93, 18 | 96.19, 6.17, 16 | −6.17 | 32 | 0.000 |
| PEP-Fine motor | 61.83, 23.59, 18 | 88.81, 16.01, 16 | −3.85 | 32 | 0.000 |
| PEP-Gross motor | 59.33, 27.86, 18 | 90.56, 7.66, 16 | −4.56 | 32 | 0.000 |
| PEP-Visual Motor Imitation | 56.11, 25.29, 18 | 93.69, 6.93, 16 | −6.05 | 32 | 0.000 |
| VABS-II-Adaptive Behaviour Composite | 75.5, 10.73, 18 | 105.28, 10.19, 18 | −8.53 | 34 | 0.000 |
| VABS-II-Communication | 76.5, 12.59, 18 | 107.28, 8.16, 18 | −8.7 | 34 | 0.000 |
| VABS-II-Daily living skills | 79.94, 11.28, 18 | 103.56, 9.28, 18 | −6.85 | 34 | 0.000 |
| VABS-II-Socialization | 74.67, 11.26, 18 | 102.89, 6.98, 18 | −9.03 | 34 | 0.000 |
| VABS-II-Motor Skills | 83.56, 10.8, 18 | 101.44, 12.15, 18 | −4.66 | 34 | 0.000 |
| VABS-II-receptive language | 10, 2.45, 18 | 16.89, 2.32, 18 | −8.65 | 34 | 0.000 |
| VABS-II-expressive language | 10.11, 2.4, 18 | 16.56, 1.5, 18 | −9.65 | 34 | 0.000 |
| VABS-II-gross motor skills | 13.83, 5.53, 18 | 14.89, 1.78, 18 | −0.77 | 34 | 0.449 |
| VABS-II-fine motor skills | 12.06, 2.58, 18 | 15.61, 2.45, 18 | −4.23 | 34 | 0.000 |
| VABS-II-interpersonal relationships | 10.06, 2.6, 18 | 15.83, 2.26, 18 | −7.12 | 34 | 0.000 |
| VABS-II-play and leisure time | 10.33, 2.03, 18 | 17, 1.68, 18 | −10.73 | 34 | 0.000 |

verified and confirmed with either the Autism Diagnosis Observation Schedule-Generic (*Lord et al., 2000*) or the Autism Diagnosis Observation Schedule, second edition (ADOS-2) (*Luyster et al., 2009*). The latter contains a toddler module that defines concern for ASD. ADOS assessments were administered and scored by experienced clinicians working at the institution and specialized in ASD identification. In order to compare scores from different modules, we transformed the ADOS-G scores into Calibrated Severity Scores (ADOS-CSS) (*Gotham et al., 2009*). For the participants that underwent the ADOS-2-toddler module, we calibrated the scores into Severity Scores (*Esler et al., 2015*). Five children under 30 months of age performed the toddler module of the ADOS-2. All scored in the moderate to severe range of concern for ASD. For all the participants younger than 3 years of age (n = 10) at the EEG acquisition, clinical diagnosis was confirmed after one year by a clinician specialized in ASD identification using the ADOS-G or ADOS-2. The mean global ADOS-CSS for the entire group of patients with ASD was 7.9 (SD = 1.6). The assessment of the participants with ASD also included the administration of additional clinical standardized tests. Adaptive behaviour was assessed using the Vineland Adaptive Behaviour Scale-II (VABS-II) (*Sparrow et al., 2005*), a standardized parent report interview. Developmental level was assessed with the Psycho-educational Profile Third Edition (PEP-3) (*Lansing et al., 2005*). See *Table 1* for characteristics of study participants. Prior to their inclusion in our research protocol, potential TD participants were initially screened for neurological/psychiatric problems and learning disabilities using a medical and developmental history questionnaire before their visit. Moreover, they underwent ADOS-G or ADOS-2 evaluations to exclude any ASD symptomatology. Fourteen controls were tested with Modules 1 or 2 and four underwent the toddler module of the ADOS-2. All TD participants had a minimal severity score of 1, except one child who had a score of 3.

## Stimuli

Stimuli consisted of two video sequences of dynamic social images without audio information of approximatively two minutes each. These videos included ecologically valid and complex naturalistic

**Table 2.** Acronyms Table

| | | |
|---|---|---|
| 1 | PreCG.L | Precentral Gyrus Left |
| 2 | PreCG.R | Precentral Gyrus Right |
| 3 | SFGdor.L | Frontal Superior Left |
| 4 | SFGdor.R | Frontal Superior Right |
| 5 | ORBsup.L | Frontal Superior Orbital Left |
| 6 | ORBsup.R | Frontal Superior Orbital Right |
| 7 | MFG.L | Frontal Middle Left |
| 8 | MFG.R | Frontal Middle Right |
| 9 | ORBmid.L | Frontal Middle Orbital Left |
| 10 | ORBmid.R | Frontal Middle Orbital Right |
| 11 | IFGoperc.L | Frontal Inferior Operculum Left |
| 12 | IFGoperc.R | Frontal Inferior Operculum Right |
| 13 | IFGtriang.L | Frontal Inferior Triangularis Left |
| 14 | IFGtriang.R | Frontal Inferior Triangularis Right |
| 15 | ORBinf.L | Frontal Inferior Orbital Left |
| 16 | ORBinf.R | Frontal Inferior Orbital Right |
| 17 | ROL.L | Rolandic Operculum Left |
| 18 | ROL.R | Rolandic Operculum Right |
| 19 | SMA.L | Supplementary Motor Area Left |
| 20 | SMA.R | Supplementary Motor Area Left |
| 21 | OLF.L | Olfactory Left |
| 22 | OLF.R | Olfactory Right |
| 23 | SFGmed.L | Frontal Superior Medial Left |
| 24 | SFGmed.R | Frontal Superior Medial Right |
| 25 | ORBsupmed.L | Frontal Medial Orbital Left |
| 26 | ORBsupmed.R | Frontal Medial Orbital Right |
| 27 | REC.L | Rectus Left |
| 28 | REC.R | Rectus Right |
| 29 | INS.L | Insula Left |
| 30 | INS.R | Insula Right |
| 31 | ACG.L | Cingulum Anterior Left |
| 32 | ACG.R | Cingulum Anterior Right |
| 33 | DCG.L | Cingulum Middle Left |
| 34 | DCG.R | Cingulum Middle Right |
| 35 | PCG.L | Cingulum Posterior Left |
| 36 | PCG.R | Cingulum Posterior Right |
| 37 | HIP.L | Hippocampus Left |
| 38 | HIP.R | Hippocampus Right |
| 39 | PHG.L | ParaHippocampal Left |
| 40 | PHG.R | ParaHippocampal Right |
| 41 | AMYG.L | Amygdala Left |
| 42 | AMYG.R | Amygdala Right |
| 43 | CAL.L | Calcarine Left |
| 44 | CAL.R | Calcarine Right |

*Table 2 continued on next page*

*Table 2 continued*

| 45 | CUN.L | Cuneus Left |
|----|-------|-------------|
| 46 | CUN.R | Cuneus Right |
| 47 | LING.L | Lingual Left |
| 48 | LING.R | Lingual Right |
| 49 | SOG.L | Occipital Superior Left |
| 50 | SOG.R | Occipital Superior Right |
| 51 | MOG.L | Occipital Middle Left |
| 52 | MOG.R | Occipital Middle Right |
| 53 | IOG.L | Occipital Inferior Left |
| 54 | IOG.R | Occipital Inferior Right |
| 55 | FFG.L | Fusiform Left |
| 56 | FFG.R | Fusiform Right |
| 57 | PoCG.L | Postcentral Left |
| 58 | PoCG.R | Postcentral Right |
| 59 | SPG.L | Parietal Superior Left |
| 60 | SPG.R | Parietal Superior Right |
| 61 | IPL.L | Parietal Inferior Left |
| 62 | IPL.R | Parietal Inferior Right |
| 63 | SMG.L | SupraMarginal Left |
| 64 | SMG.R | SupraMarginal Right |
| 65 | ANG.L | Angular Left |
| 66 | ANG.R | Angular Right |
| 67 | PCUN.L | Precuneus Left |
| 68 | PCUN.R | Precuneus Right |
| 69 | PCL.L | Paracentral Lobule Left |
| 70 | PCL.R | Paracentral Lobule Right |
| 71 | HES.L | Heschl Left |
| 72 | HES.R | Heschl Right |
| 73 | STG.L | Temporal Superior Left |
| 74 | STG.R | Temporal Superior Right |
| 75 | TPOsup.L | Temporal Pole Superior Left |
| 76 | TPOsup.R | Temporal Pole Superior Right |
| 77 | MTG.L | Temporal Middle Left |
| 78 | MTG.R | Temporal Middle Right |
| 79 | TPOmid.L | Temporal Pole Middle Left |
| 80 | TPOmid.R | Temporal Pole Middle Right |
| 81 | ITG.L | Temporal Inferior Left |
| 82 | ITG.R | Temporal Inferior Right |

dynamic images where young children practised yoga alone, imitated animal-like behaviours (behaving like a monkey or jumping like a frog), waived their arms, struck a pose, jumped, made faces or whistled (Yoga Kids 3 ; Gaiam, Boulder, Colorado, http://www.gaiam.com, created by Marsha Wenig, http://yogakids.com/). Presentation and timing of stimuli were controlled by Tobii Studio software (Sweden, http://www.tobii.com).

## Procedure and task

The experiment was conducted in a lit room at the office Médico-Pédagogique in Geneva. To familiarize the child with the procedure, the families received a kit containing a custom-made EEG replica cap and pictures illustrating the protocol two weeks prior to their first visit. Participants were seated on their parents lap in order to make them feel as secure as possible and to minimize head and body movements or alone. Once seated, the experimenter measured the circumference of the head and placed the corresponding cap on the participant's head. A couple of minutes were taken in order to allow the participants to settle into the experiment's environment and get used to the cap before starting the experiment. Following this, a five point eye-tracking calibration procedure was initiated using the Tobii system (Sweden, http://www.tobii.com). An attractive colourful object (either a kitten, a bus, a duck, a dog or a toy) was presented together with its corresponding sound on a white background and the participants had to follow the object visually. The recording and presentation of the visual stimuli started when a minimum of four calibration points were acquired for each eye. To best capture the child's attention, we first showed them an age-appropriate animated cartoon, followed by some fractals and another animated cartoon. The block ended with a film containing dynamic social images, the condition of interest in the present experiment. All participants were presented with the same visual stimuli in the same order. Following the first block, impedances were rechecked and electrodes were readjusted where needed to maintain them below 40 k*Ohm*. A second block was then acquired (animated cartoon; animated fractals; animated cartoon; second condition of interest: dynamic social images). The experimenter continuously monitored the eye-tracking to ensure children were looking at the screen. The whole experiment lasted about half an hour. We used stringent criteria and only participants with the highest data quality were kept for subsequent analysis.

## Eye-tracking measurements

Eye-tracking data were recorded with the TX300 Tobii eye-tracking system (sampling rate resolution of 300 Hz). In order to analyse and quantify differences in visual exploration between our groups, we developed a data-driven method to define dynamic norms of the exploration of the visual scenes (Kojovic et al., in preparation). First, we applied a kernel density distribution estimation (*Botev et al., 2010*) on the eye-tracking data recorded from the TD group at each time frame of the films containing dynamic complex social images to compute a normative gaze distribution pattern. Then, for each of the participants with ASD individually, we computed a deviation index from this normative gaze distribution, and this, for each single time frame separately (*Figure 6*). We averaged these values across the two films to obtain a mean Proximity Index (PI) value. This index describes for a given ASD participant, his distance from the normative gaze distribution pattern calculated on the TD group. A high index value indicates a visual behaviour approaching the visual exploration of

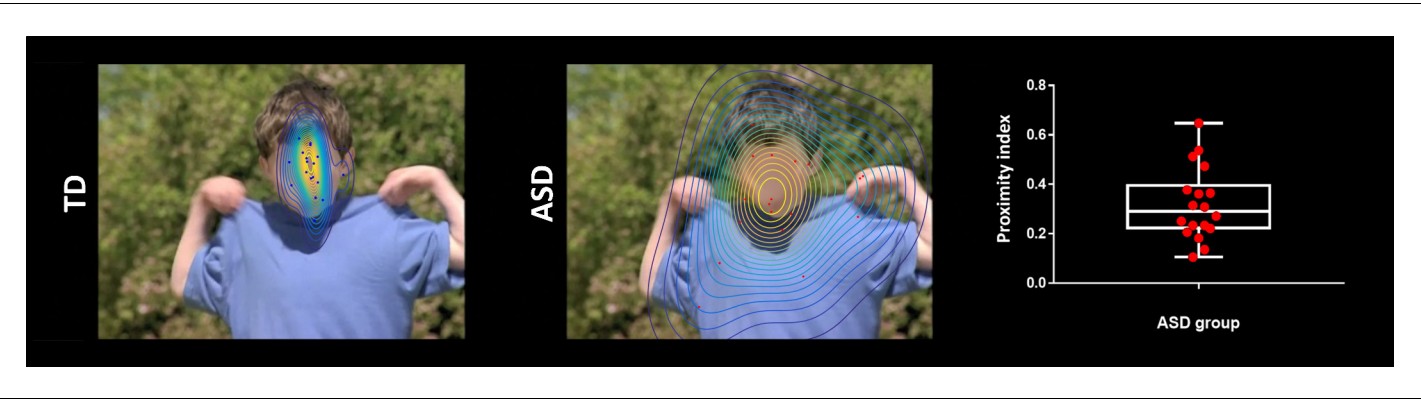

**Figure 6.** Exemplar single time frame of the normative gaze pattern for each group on one random time frame. Each dot represents the gaze position for an individual participant. The face has been blurred on purpose to preserve anonymity but was fully visible for the participants during the experiment.

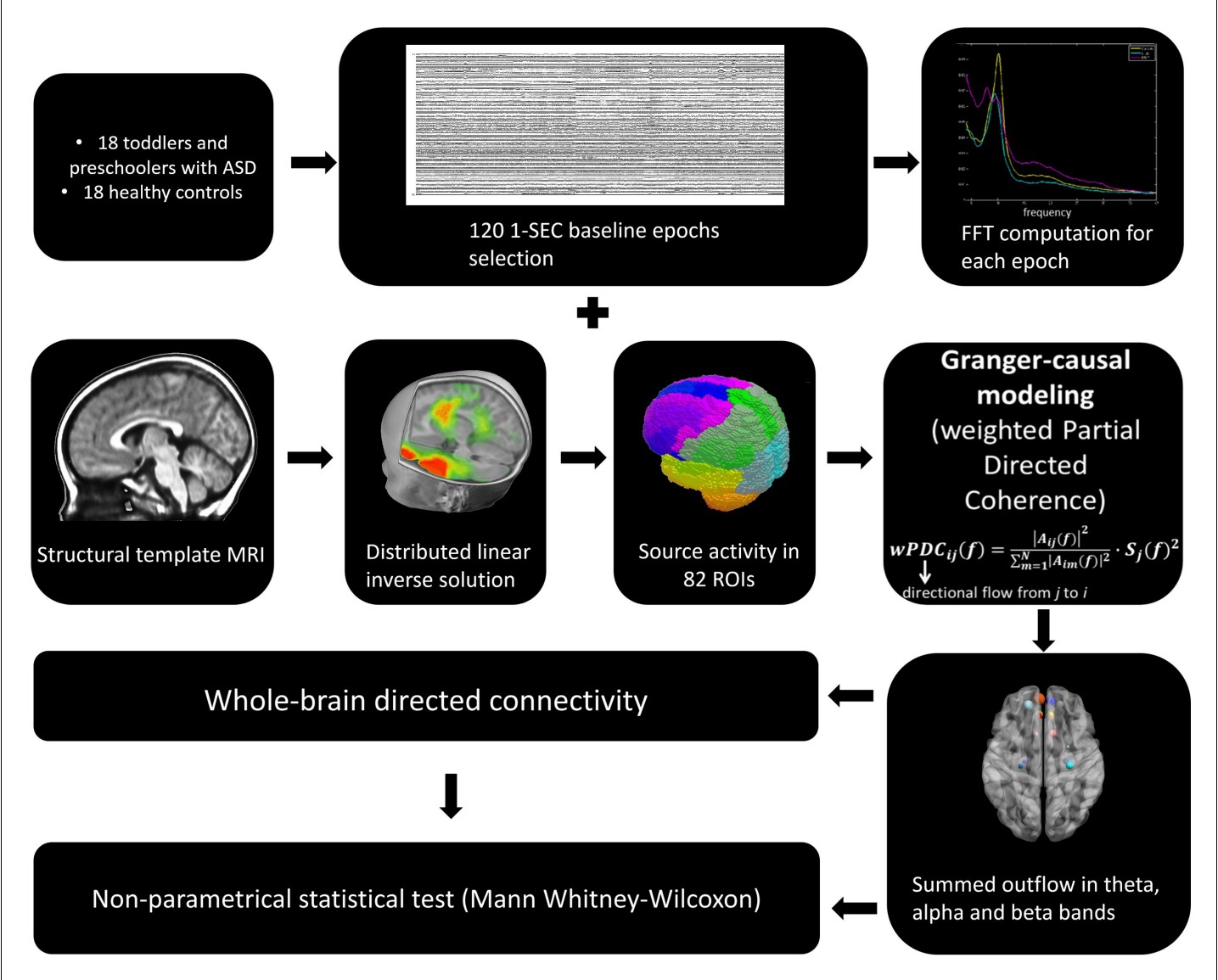

**Figure 7.** The general analysis strategy.

the TD participants (more similarity), while a low index indicates a visual behaviour deviating from the TD group (more dissimilarity).

## EEG acquisition and preprocessing

The EEG was acquired with a Hydrocel Geodesic Sensor Net (HCGSN, Electrical Geodesics, USA) with 129 scalp electrodes at a sampling frequency of 1000Hz. On-line recording was band-pass filtered at 0−100Hz using the vertex as reference. Data pre-processing was done using Matlab (Natick, MA) and Cartool (http://sites.google.com/site/cartoolcommunity/). We down-sampled the montage to a 111-channel electrode array to exclude electrodes on the cheek and the neck since those are often contaminated with artefacts. Data were filtered between 1 and 40Hz (using non-causal filtering) and a 50Hz notch filter was applied. Each file was then visually inspected by one of the three EEG experts (HFS, TAR, and RKJ) to exclude periods of movements artefacts. Periods where subjects were not looking at the screen were excluded. Independent component analysis (ICA) was performed on the data to identify and remove the components related to eye movement artefacts (eye blinks, saccades). Subsequently, channels with substantial noise were interpolated using spherical spline interpolation for each recording. Finally, the cleaned data were down-sampled to 125Hz,

recalculated against the average reference and inspected by two EEG experts (HFS and AC) to ensure that no artefacts had been missed. One hundred and twenty artefact-free epochs of 1 second per participant were included for further analysis and were considered as a minimum to ensure high enough data quality.

## Electrical Source Imaging and selection of regions of interest

The general analysis strategy is summarized in *Figure 7*. Electrical source imaging (ESI) was performed to reconstruct the sources of brain activity that gave rise to the scalp EEG field. For this, we used a toddler template head model (33–44 month) (using the Montreal Neurological Institute (MNI) brain) with consideration of skull thickness (Locally Spherical Model with Anatomical Constraints, LSMAC). 4159 solution points were equally distributed in the grey matter. We used a distributed linear inverse solution (Low Resolution Electromagnetic Tomography, LORETA [*Pascual-Marqui et al., 1994*]) to compute the 3-dimensional (3D) current source densities. We then projected this 3D dipole time-series onto the predominant dipole direction of each region of interest (ROI) across time and epochs, therefore obtaining a scalar time-series (*Coito et al., 2016a*; *Coito et al., 2015*; *Plomp et al., 2015a*; *Coito et al., 2016b*). We parcelled the grey matter in 82 ROIs based on the automated anatomical labelling (AAL) digital atlas (*Tzourio-Mazoyer et al., 2002*), after normalization to the MNI space using SPM8 (Wellcome Trust Centre for Neuroimaging, University College London, UK, www.fil.ion.ucl.ac.uk/spm). In order to reduce the dimensionality of the solution space, we considered the solution point closest to the centroid of each ROI as representative of the source activity in that ROI for further analysis. This allowed to obtain the source activity across time of 82 solution points, representative of 82 ROIs (*Coito et al., 2016b*).

## Directed functional connectivity using Granger-causality

Directed functional connectivity estimates the influence that one signal exerts onto another, facilitating the study of directional relationships between brain regions. It is commonly assessed using the concept of Granger-causality: given two signals in a process, if the knowledge of the past of one allows a better prediction of the presence of the other signal in the process, then the former signal is said to Granger-cause the latter signal (*Granger, 1969*).

Granger-causal modelling is a well-validated statistical method (*Bressler and Seth, 2011*) that has been successfully applied to estimate the strength of directed interactions between brain regions in rats using epicranial EEG (*Plomp et al., 2014*) and in non-human primates using intracranial recordings (*Brovelli et al., 2004*; *Saalmann et al., 2012*). It has also efficiently been used to study connectivity patterns in healthy humans with source imaging using EEG (*Astolfi et al., 2007*; *Hu et al., 2012*; *Plomp et al., 2015b*) and MEG (*Michalareas et al., 2016*). This approach has also effectively been applied in clinical populations to study network alterations in patients with focal epilepsy using intra-cranial recordings (*Wilke et al., 2009*; *van Mierlo et al., 2013*; *van Mierlo et al., 2011*) as well as electrical source imaging (*Ding et al., 2007*; *Coito et al., 2016a*; *Coito et al., 2015*; *Coito et al., 2016b*).

In order to have interpretable results, Granger-causality analysis should be performed using electrical source imaging rather than on electrodes measured at the scalp surface (*Schoffelen and Gross, 2009*; *Bastos et al., 2015*). Therefore, in order to estimate the directional relationships in our data, we computed the weighted Partial Directed Coherence (wPDC) (*Baccalá and Sameshima, 2001*; *Astolfi et al., 2006*; *Plomp et al., 2014*) using the 82 source signals. PDC is a multivariate approach, which considers all signals simultaneously in the same model and estimates brain connectivity in the frequency domain. It is computed using multivariate autoregressive models of a certain model order. Here, we used a model order of 5, corresponding to 40ms. The wPDC was computed for each subject and epoch and then, the average of the PDC values within subjects was taken (*Coito et al., 2016b*). The average PDC was subsequently scaled $(0 - 1)$ across ROIs and frequencies $(1 - 40 \text{ Hz})$ by subtracting the minimum power and dividing by the range. In order to weight the PDC by the spectral power (SP) of each source signal, while avoiding frequency doubling, we computed the Fast Fourier Transform (FFT) for each electrode, applied ESI to the real and imaginary part of the FFT separately and then combined them (*Coito et al., 2016a*; *Coito et al., 2015*; *Plomp et al., 2015a*; *Yuan et al., 2008*). The mean SP was obtained for each subject and scaled (that is 0–1, in the same way as PDC). For further details on the methodological approach to

compute directed functional connectivity from electrical-source imaging signals, we refer the reader to (*Coito et al., 2016b*). For each subject, we obtained a 3D connectivity matrix (ROIs x ROIs x frequency), representing the outflow from one ROI to another for each frequency. For further analysis, we reduced the connectivity matrix to 3 frequency bands: theta ($4 - 7Hz$), alpha ($8 - 12Hz$) and beta ($13 - 30Hz$), by calculating the mean connectivity value in each band. For each subject and frequency band, we computed the summed outflow as the sum of wPDC values from a given ROI to all the others. This reflects the driving importance of this ROI in the network: ROIs with high summed outflow strongly drive the activity of other ROIs. We identified the highest information transfer (summed outflow) in the theta band followed by the alpha band. Therefore, we focused our subsequent analysis on these two frequency bands. We carried out statistical comparisons of the summed outflows between subject groups using a non-parametrical statistical test (Mann − Whitney − Wilcoxon, two − tailed, $p < 0.05$). We then investigated the outflows from the ROIs that showed statistically significant summed outflow between groups to the whole brain (remaining 81 ROIs) and carried out a statistical comparison of these outflows between groups (Mann − Whitney − Wilcoxon, two − tailed, $p < 0.05$, Benjamini − Hochberg = 0.05). We correlated (Spearman − rho, two − tailed, $p < 0.05$) the summed outflow results obtained in each of the 82 ROIs with ADOS-CSS scores, with developmental scores obtained from the PEP-3, with adaptive scores obtained from VABS-II and with the PI values obtained from the eye-tracking data. In all cases, correlation p-values were Benjamini-Hochberg-corrected for multiple testing with $p = 0.05$. Connectivity computations were performed in Matlab. *Figures 1*, *2* and *3*, *Figure 3—figure supplement 2*, *Figure 3—figure supplement 2*, *Figures 4* and *5* were produced using the BrainNet Viewer toolbox (*Xia et al., 2013*).

## Acknowledgements

The authors would like to express their gratitude to all the families who took part in this research. This research is supported by a grant from the National Centre of Competence in Research (NCCR) 'SYNAPSY-The Synaptic Bases of Mental Diseases' financed by the Swiss National Science Foundation (number: 51AU40-125759) and by private funding by the Fondation Pole Autisme (http://www. pole-autisme.ch). This work was further supported by a SNSF grant to MS. (number: 163859), as well as SNSF grants number 320030–159705 to CMM and number 169198 and CRSII5170873 to SV. RKJ received individual support from a Marie Curie fellowship, which received funding from the European Union Seventh Framework Programme (FP7:2007–2013) under grant agreement number 267171.

## Additional information

### Funding

| Funder | Grant reference number | Author |
|---|---|---|
| National Centre of Competence in Research | 51AU40-125759 | Stephan Eliez<br>Christoph M Michel<br>Marie Schaer |
| Swiss National Science Foundation | 163859 | Marie Schaer |
| Swiss National Science Foundation | 320030-159705 | Christoph M Michel |
| Swiss National Science Foundation | 169198 | Serge Vulliemoz |
| Swiss National Science Foundation | CRSII5170873 | Serge Vulliemoz |
| Marie Curie Fellowship | 267171 | Reem Kais Jan |
| Seventh Framework Programme | 267171 | Reem Kais Jan |
| Fondation Pôle Autisme | | Stephan Eliez<br>Marie Schaer |

The funders had no role in study design, data collection and interpretation, or the decision to submit the work for publication.

## Author contributions

Holger Franz Sperdin, Conceptualization, Formal analysis, Investigation, Visualization, Methodology, Writing—original draft, Writing—review and editing; Ana Coito, Conceptualization, Software, Formal analysis, Validation, Investigation, Visualization, Methodology, Writing—original draft, Writing—review and editing; Nada Kojovic, Software, Formal analysis, Investigation, Visualization; Tonia Anahi Rihs, Conceptualization, Investigation, Writing—review and editing; Reem Kais Jan, Martina Franchini, Investigation, Writing—review and editing; Gijs Plomp, Software, Methodology, Writing—review and editing; Serge Vulliemoz, Supervision, Funding acquisition, Writing—review and editing; Stephan Eliez, Resources, Supervision, Funding acquisition; Christoph Martin Michel, Conceptualization, Resources, Software, Supervision, Funding acquisition, Methodology, Project administration, Writing—review and editing; Marie Schaer, Conceptualization, Resources, Data curation, Software, Supervision, Funding acquisition, Investigation, Methodology, Project administration, Writing—review and editing

## Author ORCIDs

Holger Franz Sperdin (iD) http://orcid.org/0000-0002-3438-1572
Tonia Anahi Rihs (iD) http://orcid.org/0000-0002-6290-7439
Gijs Plomp (iD) http://orcid.org/0000-0002-9883-3371

## Ethics

Human subjects: Prior to the experiments, all the procedures were approved by the Ethics Committee of the Faculty of Medicine of the University of Geneva Hospital in accordance with the ethical standards proclaimed in the Declaration of Helsinki. For all participants, an interview over the phone and a medical developmental history questionnaire were completed before their initial visit. All participants' parents gave their informed consent prior to inclusion in the study.

## Decision letter and Author response

Decision letter https://doi.org/10.7554/eLife.31670.sa1
Author response https://doi.org/10.7554/eLife.31670.sa2

# Additional files

## Supplementary files

• Transparent reporting form

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
