## [Decision Letter]

Thank you for submitting your article "Early alterations of social brain networks in young children with autism" for consideration by *eLife*. Your article has been favorably evaluated by Sabine Kastner (Senior Editor) and three reviewers, one of whom is a member of our Board of Reviewing Editors. The reviewers have opted to remain anonymous.

The reviewers have discussed the reviews with one another and the Reviewing Editor has drafted this decision to help you prepare a revised submission.

Summary:

This manuscript used multisite EEG, source localization, and functional connectivity analyses to detect differences in brain network activity in 3-year old children with autism viewing a social video, providing, according to the authors, the first functional connectivity analysis of the toddler brain activated by social stimuli.

Characterizing EEG activity in autistic children is critical for understanding the underlying cause. Ultimately it is the electrical brain activity that can provide the important mechanistic insights that lead to the observable behavioral phenotype. Thus, this study has the potential to become an important contribution to the field of autism.

Indeed, the reviewers congratulate the authors for the novelty of this approach and bringing it to the study of humans with neurodevelopmental disorders. The reviewers suggest that the work could somehow be re-framed to bridge the gap between Astolfi et al. (2006) with Coito (2016).

Essential revisions:

1) While the reviewers admire the work performed in this study, and agree that EEG-based directional functional connectivity measures constitute an important and promising approach to study autism, the sole reliance on the Granger method raises concerns. This skepticism is based on the lack of prior literature, and the findings presented here – chiefly much connectivity activity seen at the hippocampus, a structure that generates little to the EEG. Probably the majority of EEG researchers, the reviewers among them, who have attempted directed functional connectivity estimation at the scalp level find the method spurious. For example, connectivity at electrode pairs can always be computed, but validation of the direction and weight seems unreliable, for example estimated at nearest neighbor electrodes, yield irregular directions and weights.

We agree, that intracranial EEG and LFP in animals show interpretable Granger causality and there are many fine publications on this. However, to the best of our knowledge there are no comparable papers based on scalp EEG. The signal may be too noisy for the Granger method, and this is why it has not been widely applied. We understand that the authors are not examining EEG at electrodes, but they examine connectivity in source space. Methodology here appears conservative with LORETA technique and appropriate MNI template model. This suggests the accuracy of source reconstruction is probably good – to the extent that reconstruction will be good in three-year-children.

As for the revision: To increase the confidence in the results presented in this study, additional work would have to be done to demonstrate stable Granger patterns. This could be done in any number of ways, and we leave it up to the authors to find the best way: One possible approach might be done at the voxel level. Neighboring voxels should show similar directed functional connectivity. This could be demonstrated and Granger details documented in revision, such that future research can replicate the study.

2) Analyses were restricted to theta due to the finding that less "summed outflow" occurred in the α and β bands. Although I recognize that the choice was guided by practicalities of analysis as well as the understanding that theta relates to general activation and memory formation, the figures seem to indicate in some sites less α activity with ASD, relating perhaps to a less relaxed mental state. Whether this interpretation of the α data is correct or not, this comment gets to the issue of whether the paper achieves its greatest potential by restricting analysis to theta. The paper could have a much greater impact if there was a contrast between directed coherence in the theta and α bands.

3) Contrasting α and theta could help with a significant limitation of the current analysis which produces a list of region-to-region changes with ASD which, in toto, demonstrate details of what is already known, namely that brain activity is different between ASD and TD children. The α and theta contrast might provide functional meaning to the changes, which otherwise are difficult to interpret.

4) As an example of a difficult-to-interpret finding using theta alone, there is a strong correlation between output of the left Heschl's gyrus and fine motor skills. This would seem to imply an auditory receptive language component in motor skill learning, not necessarily performance as suggested, and could be a general property upstream of motor function. Certainly, α band activity in motor cortex has been related to motor performance per se.

5) A compelling argument is made that greater summed outflow (re: TD) in some brain areas for some ASD individuals relates to better gaze performance, and thus may be a compensatory mechanism for ASD. This seems to be the strongest aspect of this manuscript and deserves greater attention. It would be interesting to report whether the individuals that compensated in this way had lower ADOS scores, or better socialization, as compared to those that did not, and, more importantly, whether (or not) such compensations were uniform across functional domains. Does each individual that shows such compensation have an overall similar (or different) compensatory architecture in either their summed outflow from other ROIs or their overall functional connectivity? If so, what are the implications for therapy?

---

## [Author Response]

Essential revisions:1) While the reviewers admire the work performed in this study, and agree that EEG-based directional functional connectivity measures constitute an important and promising approach to study autism, the sole reliance on the Granger method raises concerns. This skepticism is based on the lack of prior literature, and the findings presented here – chiefly much connectivity activity seen at the hippocampus, a structure that generates little to the EEG. Probably the majority of EEG researchers, the reviewers among them, who have attempted directed functional connectivity estimation at the scalp level find the method spurious. For example, connectivity at electrode pairs can always be computed, but validation of the direction and weight seems unreliable, for example estimated at nearest neighbor electrodes, yield irregular directions and weights.We agree, that intracranial EEG and LFP in animals show interpretable Granger causality and there are many fine publications on this. However, to the best of our knowledge there are no comparable papers based on scalp EEG. The signal may be too noisy for the Granger method, and this is why it has not been widely applied. We understand that the authors are not examining EEG at electrodes, but they examine connectivity in source space. Methodology here appears conservative with LORETA technique and appropriate MNI template model. This suggests the accuracy of source reconstruction is probably good – to the extent that reconstruction will be good in three-year-children.

We would like to gratefully thank the reviewers for their summary and positive appraisal of our manuscript. As we understand, the first concern is that the sole reliance on the Granger method raises concern because of the lack of prior literature. As the reviewers further mention, there are several publications based on intracranial EEG data and LFP in animals showing interpretable Granger causality but no comparable papers based on scalp EEG. In our experience too, the SNR is higher in recordings from animal models, and this certainly helps for inferring directed functional connections. We would hasten to note, however, that comparable papers using source imaging based on EEG recordings in both clinical and healthy populations have recently been published, by our group as well as others. We now make this point more clear in the manuscript and have added the following parts in the Introduction and in the Materials and method sections:

Introduction: “This approach is being increasingly and successfully applied to study connectivity alterations using intra-cranial recordings (Wilke, van Drongelen et al. 2009, van Mierlo, Carrette et al. 2011, van Mierlo, Carrette et al. 2013) as well as source imaging based on EEG recordings in clinical populations (Ding, Worrell et al. 2007, Coito, Plomp et al. 2015, Coito, Genetti et al. 2016, Coito, Michel et al. 2016) and in healthy controls (Astolfi, Cincotti et al. 2007, Hu, Zhang et al. 2012, Plomp, Hervais-Adelman et al. 2015).”

Materials and methods: “Directed functional connectivity estimates the influence that one signal exerts onto another, facilitating the study of directional relationships between brain regions. […] In order to have interpretable results, Granger-causality analysis should be performed using electrical source imaging rather than on electrodes measured at the scalp surface (Schoffelen and Gross 2009, Bastos, Vezoli et al. 2015).”

As for the revision: To increase the confidence in the results presented in this study, additional work would have to be done to demonstrate stable Granger patterns. This could be done in any number of ways, and we leave it up to the authors to find the best way: One possible approach might be done at the voxel level. Neighboring voxels should show similar directed functional connectivity. This could be demonstrated and Granger details documented in revision, such that future research can replicate the study.

We thank the reviewers for highlighting this second concern about the stability of the Granger Causality patterns and appreciate the ideas offered for improving the manuscript.

As the reviewers mention, Granger-causality should be performed using electrical source imaging as we did in the present experiment, rather than on electrodes measured at the scalp surface. We adopted a rather conservative methodology with LORETA technique and appropriate MNI template model suggesting accurate source reconstruction. Furthermore, the results have been corrected for multiple testing using the Benjamini-Hochberg procedure (BH step-up procedure) to guard against spurious effects. Also, the data-driven approach reveals high information transfer in physiologically relevant frequency bands given the young age of our participants. The suggestion as proposed by the reviewers here, namely to demonstrate whether neighboring voxels (i.e. solutions points) show similar Granger patterns would require doing a very high resolution reanalysis for each of the solution points (4159 solution points), which is not possible with the multivariate methods we use. Moreover, in order to reduce the dimensionality of the solution space, we considered the solution point closest to the centroid of each ROI. Thus, for a single ROI only a single solution point is representative of the source activity in this ROI (82 solution points, representative of the 82 ROIs). However, in order to increase the confidence in the results, we now include boxplots with summed outflow values for each of the significant ROIs comparing our two groups and this for all the significant ROIs (see new Figure 2B for the theta band and Figure 2C for α band). As can be seen from these boxplots, systematic differences between the groups are present for all the significant ROIs and for both frequency bands suggesting consistent (stable) Granger patterns across participants. If Granger patterns were spurious (that is random), then these systematic differences between the groups would not be obtained. We have also added boxplots for the ORBsup.R seed regions in the theta band showing outflow values toward each of the selected target regions for the ASD group and for the TD group as Figure 3—figure supplement 1 and Figure 3—figure supplement 2, respectively.

2) Analyses were restricted to theta due to the finding that less "summed outflow" occurred in the α and β bands. Although I recognize that the choice was guided by practicalities of analysis as well as the understanding that theta relates to general activation and memory formation, the figures seem to indicate in some sites less α activity with ASD, relating perhaps to a less relaxed mental state. Whether this interpretation of the α data is correct or not, this comment gets to the issue of whether the paper achieves its greatest potential by restricting analysis to theta. The paper could have a much greater impact if there was a contrast between directed coherence in the theta and α bands.

We thank the reviewer for this observation. We have now performed the analysis also for the α band (8-12Hz), which we now include in the revised manuscript. The new analyses revealed α specific network atypicalities, which we now include and discuss in detail in the manuscript. Please see also our reply in points (3) and (4) below for further details.

3) Contrasting α and theta could help with a significant limitation of the current analysis which produces a list of region-to-region changes with ASD which, in toto, demonstrate details of what is already known, namely that brain activity is different between ASD and TD children. The α and theta contrast might provide functional meaning to the changes, which otherwise are difficult to interpret.

We now contrast theta and α. The global driving in the theta and α bands did not differ between groups (theta: df=34, t=0.6201, p = 0.536; α: df=34, t=0.1736, p = 0.8632). Driving in the theta band was higher compared to the driving in the α band in both groups (ASD: df=17, t=11.86, p < 0.0001; TD: df=17, t=8.025, p < 0.0001) (see new Figure 1B). While we found six ROIs in the theta band with an altered driving, in the α band we found three ROIs that had a different driving in the ASD group compared to the TD group. The right orbital part of the middle frontal gyrus (Ws = 262, z = -2.246, p= 0.024, r = -0.374) and the left cuneus (Ws = 265, z = -2.151, p= 0.031, r = -0.358) had a higher driving and the right STG had a weaker driving (Ws = 265, z = -2.151, p= 0.031, r = -0.358) (see new Figure 2A). The region-to-region directed functional connectivity analysis for the α band revealed stronger connections from the right orbital part of the middle frontal gyrus and the left cuneus whereas the right STG had weaker connection in the toddlers and preschoolers with ASD compared to their TD peers (Mann-Whitney-Wilcoxon, two-tailed, p<0.05, Benjamini-Hochberg = 0.05). All three significant ROIs in α had different network patterns in the toddlers and preschoolers with ASD compared to their TD peer (see new Figure 4). We now display the connections from all the seed regions in both bands and for both groups in new Figure 3 (theta) and new Figure 4 (α) (see Results and Discussion sections in the manuscript).

4) As an example of a difficult-to-interpret finding using theta alone, there is a strong correlation between output of the left Heschl's gyrus and fine motor skills. This would seem to imply an auditory receptive language component in motor skill learning, not necessarily performance as suggested, and could be a general property upstream of motor function. Certainly, α band activity in motor cortex has been related to motor performance per se.

We thank the reviewer for this observation. We have now explored associations between the summed outflow in α band with gaze performance and clinical and behavioral phenotype as we did for the theta band. For the α band, we only found two significant correlations after FDR correction: higher summed outflow within the left hippocampus and the left rolandic operculum were positively related to better fine (rs = 0.736,N = 18, p = 0.0005, two-tailed, <0.05; Benjamini-Hochberg = 0.05) and gross motor skills (rs = 0.737,N = 18, p = 0.0005, two-tailed, <0.05; Benjamini-Hochberg = 0.05) as measured by the PEP-3 (see new Figure 5). Thus, higher summed outflow in α band within these areas was related to better motor performance per se. We didn’t find any other relationships between the summed outflow in α and gaze performance, the ADOS-2 severity scores, adaptive behavior scores using the VABS-II or items form the developmental level assessed with the PEP-3 suggesting that increased driving in the α band was confined to better motor performance (see Results and Discussion sections in the manuscript).

5) A compelling argument is made that greater summed outflow (re: TD) in some brain areas for some ASD individuals relates to better gaze performance, and thus may be a compensatory mechanism for ASD. This seems to be the strongest aspect of this manuscript and deserves greater attention. It would be interesting to report whether the individuals that compensated in this way had lower ADOS scores, or better socialization, as compared to those that did not, and, more importantly, whether (or not) such compensations were uniform across functional domains. Does each individual that shows such compensation have an overall similar (or different) compensatory architecture in either their summed outflow from other ROIs or their overall functional connectivity? If so, what are the implications for therapy?

Following the reviewer’s remark, we examined the possibility that toddlers with ASD with better gaze performance are the ones with lower autistic severity, better adaptive functioning and higher cognitive developmental scores. To this purpose, we correlated the Proximity Index (i.e. gaze performance) with the global severity of autistic symptoms as measured by the ADOS-2 calibrated severity score, the PEP-3 scores in the developmental areas of cognitive verbal/preverbal, expressive and receptive language, fine and gross motor skills and visual motor imitation skills and with VABS-II standardized Adaptive Behavior Composite scores as well as the four domain scores: Communication, Daily Living Skills, Socialization, and Motor Skills (see Table 1). We didn’t find any significant correlation between the PI and the overall severity of autistic symptoms as measured with the ADOS-2 severity score. However, we found that the toddlers with ASD with a better gaze performance had better adaptive functioning as measured with the parent-reported scale of the VABS-II (total score), that was driven by improved global and fine motor skills and better development of interpersonal relationships. In terms of cognitive development, the children with better gaze performances also showed better visual motor imitation skills as measured by the PEP-3. This results reflect the fact that the toddlers with ASD who have better gaze performance, are the ones who have both less clinical impairments and greater summed outflow. Whether or not this reflects a compensatory mechanism remains an important question to be solved. Our laboratory has indeed been interested over the last years by the potential to use gaze performance measured with eye-tracking to reflect clinical phenotype or help predict clinical outcome over time, so that the above-reported results that children with better gaze performance are the ones with better adaptive and cognitive functioning comes as no surprise to us. In a first cross-sectional study, similarly to the correlation between gaze performance and clinical severity reported above, we found that toddlers with ASD who spent more time exploring dynamic social images compared to geometrical forms were those who had more joint attention behaviors and improved communication skills (Franchini, Glaser et al., 2017). In a second longitudinal eye-tracking experiment, we found that toddlers with ASD who preferred social stimuli in comparison to geometric shapes at the age of 3 developed better outcome in terms of decreased symptom severity a year later (Franchini, de Wilde et al., 2016). The present study suggests that different patterns of brain activation might be associated with different clinical outcomes, but a longitudinal design with EEG and clinical assessments at both time points would be needed to fully to answer this question.

Regarding the implications for therapy, the reviewer’s question is important, but again we think that the current results might be too preliminary to support changes in existing therapeutic interventions. Nowadays, state of the art therapies for children with autism are early and intensive behavioral therapies that focus on the development of reciprocal interactions through play-based exchanges that increase the motivation of the child to orient to social cues. One of such approach is the Early Start Denver Model (Dawson et al., 2010) which is entirely based on the idea that intervening early, helping children orient to social stimuli, may partially restore aberrant trajectories of brain development in affected children. In that sense, we would expect that better gaze performance, which could be the results of therapeutic interventions or compensation by the child himself, is reflected by changes in brain activity such as greater summed outflow. Again, only a proper longitudinal design would allow to fully answer this question, and no published study to date has combined eye-tracking and EEG in a longitudinal design. The present experiment is a first step towards that direction.

To address the reviewer’s questions, we have edited the manuscript in the Results and Discussion sections. We now write:

Results: “Finally, we explored associations between gaze performance with developmental scores obtained from the PEP-3 and with adaptive scores obtained from the VABS-II (D’Agostino-Pearson omnibus normality test, K2, p<0.05; Pearson's r, two-tailed, p<0.05). […] We also found that these children with a better gaze performance had better visual motor imitation skills (K2 = 2.671, p = 0.263; r =.534, p = 0.022) as measured by the PEP-3.”

Discussion: “While to the best of our knowledge, there is currently no other relevant experimental data that addresses this question, we speculate that the overall hyper-driving from these relevant brain regions might be a mechanism to compensate for atypical development of the brain’s circuitry over time as higher directed functional connectivity was related improved socialization, motor behaviours and better visual exploration of dynamic social images. […] The present experiment is a first step towards that direction.”